# Nature Exposure and Positive Body Image: A Cross–Sectional Study Examining the Mediating Roles of Physical Activity, Autonomous Motivation, Connectedness to Nature, and Perceived Restorativeness

**DOI:** 10.3390/ijerph182212246

**Published:** 2021-11-22

**Authors:** Migle Baceviciene, Rasa Jankauskiene, Viren Swami

**Affiliations:** 1Department of Physical and Social Education, Lithuanian Sports University, LT-44221 Kaunas, Lithuania; 2Institute of Sport Science and Innovations, Lithuanian Sports University, LT-44221 Kaunas, Lithuania; rasa.jankauskiene@lsu.lt; 3School of Psychology and Sport Science, Anglia Ruskin University, Cambridge CB1 1PT, UK; viren.swami@aru.ac.uk; 4Centre for Psychological Medicine, Perdana University, Kuala Lumpur 50490, Malaysia

**Keywords:** nature exposure, body appreciation, physical activity in nature, self-determination, restoration, connectedness to nature

## Abstract

Research shows that nature exposure is directly and indirectly associated with more positive body image, which is an important facet of well-being more generally. In this study, we tested the mediating roles of physical activity in nature, perceived restoration in nature, autonomous motivation, and connectedness to nature in explaining the association between nature exposure and positive body image. An online sample of 924 Lithuanian adults (age M = 40.0 years, 73.6% women) completed a survey that included the Nature Exposure (NE) Scale, the Body Appreciation Scale-2, a measure of frequency of physical activity in nature (PAN), the Behavioral Regulation in Exercise Questionnaire-2, the Restoration Outcome Scale, and the Connectedness to Nature Scale. Path analysis was conducted to examine hypothesized direct and indirect effects. Results showed that both greater NE (B = 0.564, SE = 0.057, *p* < 0.001) and autonomy in exercise motivation (B = 0.039, SE = 0.006, *p* < 0.001) were associated with more frequent PAN. Direct effects from exercise autonomy to nature restorativeness (B = 0.017, SE = 0.006, *p* = 0.004) and body appreciation (B = 0.041, SE = 0.004, *p* < 0.001) were observed. Associations were also found between connectedness to nature and body appreciation (B = 0.166, SE = 0.040, *p* < 0.001), nature restorativeness and body appreciation (B = 0.075, SE = 0.019, *p* < 0.001), and frequency of PAN and body appreciation (B = 0.064, SE = 0.019, *p* < 0.001). PAN mediated the relationship between NE and body appreciation. The final model was invariant across place of residence (urban vs. rural) and gender. Including self-determined physical activity in nature may increase the effectiveness of intervention programs aimed at promoting more positive body image.

## 1. Introduction

### 1.1. Natural Environments and Well-Being

Much research provides robust evidence of the salutogenic effects of natural environments on human health and well-being [1,2,3,4,5]. Nature refers to “physical features and processes of nonhuman origin that people usually perceive, including ‘living nature’ of flora and fauna, with still and running water, qualities of air and weather, and the landscapes that comprise these and show the influence of geological processes” [5]. The term “nature” is used interchangeably with “natural environment” because their meanings overlap substantially, and generally refers to the continuum of environments from wild nature to designed green spaces [6].

Nature exposure is associated with improvements in physiological health through enhanced physical activity, decreased blood pressure, enhanced immune system resources, and reduced stress [7]. Psychological benefits include improved cognitive functioning, lower rates of depression and anxiety, and higher self-esteem, subjective vitality, quality of sleep, and happiness [1,8,9,10,11,12]. Two frameworks have been used to explain the associations between exposure to nature and its outcomes on human health and well-being: Psychophysiological Stress Recovery Theory (SRT) [13] and Attention Restoration Theory (ART) [14,15]. SRT suggests that human beings have an evolutionary preference for surroundings with depth, complexity, and structure and that exposure to such environments reduces stress by enhancing positive emotions, restricting negative thoughts, and supporting the parasympathetic nervous system [13]. ART meanwhile proposes that restorative environments provide human beings with opportunities to rest inhibitory mechanisms on which attention depends and therefore facilitate better recovery from mental fatigue [14].

### 1.2. Natural Environments and Positive Body Image

In addition, cross-sectional and experimental research shows that nature exposure is directly and indirectly associated with more positive body image [10,16,17,18,19], an important facet of mental health more generally [20]. Positive body image refers to a love and respect for the body, appreciation of the uniqueness of one’s body, acceptance of the body including those aspects that do not meet stereotypical beauty ideals, appreciation of the body’s functionality, and acceptance of body-protective behaviors [21,22]. Positive body image is not the opposite or the absence of the negative body image; rather, it is a multi-faceted construct related to positive embodiment, eudaimonic well-being, quality of life, higher self-esteem, mindfulness, and positive health-related behaviors [23,24] independently of negative body image. Based on SRT and ART, it has been hypothesized that the natural environment provides humans with opportunities to physically and mentally distance themselves from heavily appearance-focused societal contexts and helps to mitigate against negative thoughts and feelings related to body appearance [17,19]. Evidence from experimental studies suggests that allotment gardening, spending time in nature, or exposure to simulated (i.e., images and films of) natural environments are associated with an elevated state of positive body image [17,18,25,26,27].

Although direct relationships between nature exposure and body image outcomes have been postulated and documented [26], it is also likely that mediatory pathways exist. Indeed, the cross-sectional direct association between nature exposure and positive body image is generally moderate (rs~0.30) [19,28], which hints at the likelihood of a multiplicity of pathways involving direct and indirect relationships between nature exposure and positive body image. Consistent with this suggestion, a number of previous studies have examined mediatory mechanisms and have generally provided supporting evidence. Thus, studies have variously shown that connectedness to nature, self-compassion, and trait mindfulness all mediate (both serially and in parallel) the relationship between nature exposure and positive body image [16,19,28]. Beyond these constructs, however, there is scope to further understand mediatory pathways.

For instance, one broad category of variables that may act as a mediator is physical activity. According to the developmental theory of embodiment [29], positive embodiment can be supported through engaging in joyful and pleasant physical activities that emphasize body functionality [30] and subjective experience of the body (feeling comfortable and connected to the body, and respecting the body and its feelings, rather than judging what it looks like from an observer’s perspective). Thus, it might be hypothesized that various physical activities in the natural environment, in addition to related attitudes and cognitions, may increase positive embodiment and help to shift humans’ attention from aesthetic (i.e., body appearance) to functionality-related (i.e., vitality, physical fitness) body features [17,29]. However, to date, physical activity in nature and associated constructs have not been tested as mediators of the relationship between nature exposure and positive body image. Thus, one of the objectives of the present study is to provide more knowledge on this issue.

### 1.3. Natural Environments and Physical Activity in Nature

Three broad activity domains have been identified as physical activity in nature: work, active transport (walking and cycling), and leisure recreation (taking part in various sports and recreational activities) [5]. A recent review concluded that associations between nature exposure and physical exposure are equivocal, possibly because of the wide range of measures and methods used to measure access to the natural environment and physical activity [31]. Physical activity is known to be an important mediator of associations between nature exposure, general well-being, and mental health [5,32]. Studies focusing on physical activity in nature suggest that physical activity undertaken in natural environments provides greater mental health benefits (relaxation, stress reduction, nature enjoyment) than physical activity in indoor or other outdoor settings [33]. The framework of ecological dynamics has been used to explain why exercisers feel better after performing the same exercise in natural environments than in indoor environments [34]. Specifically, it is claimed that the unique benefits of nature-based exercise are centered on notions of affordances and variability of nature. Nature affordances are less constrained than manufactured affordances in gyms or sports clubs. Further, the variability of natural environments solicits immersive interactions and attention. Acting in natural environments, due to their variability, demands the holistic (cognitive and emotional) involvement of individuals [34]. Finally, nature-based physical activity provides opportunities for developing expertise to deal with unpredictable and challenging situations, and invitations of nature “for immersive, embodied engagement of the individual” [34]. Based on these arguments and the theory of positive embodiment [29], it is reasonable to assume that physical activity in nature may provide an indirect positive body image-enhancing effect.

### 1.4. Physical Activity in Nature, Autonomous Motivation, and Positive Body Image

Participation in sport and physical activity is generally associated with more positive body image [35,36]. Nevertheless, evidence shows that physical activity goals and quality of motivation are important mediators between these constructs. Findings based on self-determination theory (SDT) [37] suggest that body image improvement-related exercise goals, such as appearance improvement and weight control, are more strongly associated with controlled motivation regulation. In contrast, exercising for enjoyment, pleasure, health improvement, or socialization is associated with autonomous or self-determined intrinsic motivation regulation forms [38,39,40]. Further, evidence shows that, although higher exercise frequency is associated with greater positive body image, high levels of controlling exercise goals such as appearance improvement weaken this relationship [41]. Other studies have shown that positive body image (body appreciation) predicts intrinsic physical activity motivation [42].

SDT is an organismic theory and assumes that human beings naturally develop in the direction of increasing adaptation, integration, and coherence, where possible [43]. According to SDT, intrinsic motivation has a positive impact on psychological health and well-being, because it helps to fulfil three basic human needs—autonomy, relatedness, and competence (ARC) [43]. Recent evidence suggests that nature exposure and connectedness with nature may satisfy the psychological need of relatedness providing the possibility for non-human forms or relatedness [44,45]. However, there is ongoing discussion about the basic human needs and ARC might be extended adding new basic needs, such as “need for novelty” and/or “novelty–variability” [43]. If novelty–variability is truly a basic human need, it may also be fulfilled by physical activity in nature, because better restoration outcomes when exercising in nature are associated with affordances of nature variability as discussed previously [34].

There is some evidence that experiencing nature may be perceived as an important goal for physical activity. A study by [46] demonstrated that people exercising in nature are not driven by controlling body-oriented motives in comparison to sports and gym-based exercisers. Motivation to engage in physical activity in nature involves focusing on environmental factors, such as natural surroundings, rather than factors such as appearance enhancement [46]. People exercising in nature were also more likely to mention convenience and the possibility of self-regulating the intensity of physical activity as important motivational goals. In a previous study, it was found that internal exercise motivation was a mediator between the self-reported availability of natural environments in residential areas and the frequency of moderate to vigorous physical activity. Thus, it was found that more internally motivated people reported that nature environments were “closer” compared to those who are externally motivated [47]. Thus, we assume that nature exposure and physical activity in nature is associated with intrinsic-oriented physical activity motivation that helps to enhance well-being by fulfilling basic human needs (ARC and possibly “the need for novelty—variability”). Therefore, in the present study, we also tested exercise motivation as the mediator between nature exposure and physical activity in nature.

### 1.5. Natural Environments, Connectedness to Nature, Restoration in Nature, Positive Body Image, and Physical Activity

The biophilia hypothesis [48] states that human beings have an innate need to be around other living things because the human–nature relationship is driven by biological evolution. However, evidence suggests that people have different levels of connectedness to nature [49,50,51]. Connectedness to nature has been defined as a self-perceived relationship between the self and the natural environment [49,52]. Research demonstrates that connectedness to nature is associated with contact with nature, ecological behaviours, satisfaction with life, environmental identity, and eudaimonic well-being, and negatively with consumerism [49,50,51]. Connectedness to nature may help shift attention away from appearance concerns onto more holistic embodying experiences [53], and may also help to promote an attitude of ecocentric connections where humanity is seen as part of the web of life [54]. Evidence shows that nature connectedness is directly associated with positive body image [16] and mediates the associations between nature exposure and positive body image [19]. In the present study, we tested connectedness to nature as a mediator between nature exposure and positive body image, aiming to provide more empirical data on this issue.

Evidence exists that natural environments are perceived to be more restorative than urban environments [55]. The ART states that the modern world places demands on humans’ cognitive and emotional systems for which they are not necessarily well adapted, and those systems have finite resources that are depleted by urban environments [14,56]. Therefore, based on SRT and ART [13,14], nature exposure can help to restore those systems, because it places few demands on cognitive and emotional systems, decreases negative stress-related emotions, and increases positive emotions. Nature captures people’s attention, allows the executive system regulating attention to rest, and enables negative emotions and thoughts to be replaced by positive ones. Therefore, as discussed previously, visiting nature may help to decrease negative body image concerns and related emotions, and increase positive body image. However, nature restorativeness has not been tested as a mediator between nature exposure and positive body image; therefore, one of the objectives of the present study was to do so.

Although physical activity in nature is experienced to be more restorative compared to indoor physical activities [57], less is known about the associations between the connectedness to nature and physical activity. One recent study demonstrated that connectedness to nature was associated with the physical fitness goals for exercising [58]. There is some evidence that outdoor exercisers report greater relatedness to nature experience [59]. However, it is important to have more empirical data on this topic.

### 1.6. The Present Study

To date, the majority of studies analyzing interrelationships between nature exposure and positive body image have been conducted in Western populations, and it is unclear to what extent these findings will generalize to populations from other countries [60]. Thus, the present study also aimed to replicate previous findings in Lithuanian adults and to add new knowledge on the topic. Specifically, we tested the mediating role of physical activity in nature and autonomous motivation in the associations between nature exposure and positive body image (operationalized as body appreciation). Based on previous findings, we predicted that physical activity in nature and autonomous motivation would mediate the association between nature exposure and positive body image. Second, aiming to replicate findings in Western populations [16,19,28], we sought to extend current knowledge by testing the path model of nature exposure and positive body image including measures of perceived restoration and connectedness to nature. We predicted that nature restoration would mediate relationships between nature exposure and positive body image on the one hand and connectedness to nature on the other. The hypothesized model is provided in Figure 1. For exploratory purposes, we also assessed the extent to which the final, derived model was invariant across urban and rural residents, and across gender, as these are elements that scholars have identified as being important to test [26].

## 2. Materials and Methods

### 2.1. Procedure

All data were collected via an online survey between January and April 2021. Participants were recruited using a non-probabilistic sampling method. Inclusion criteria were set only for age (18 years and over) and language spoken (Lithuanian). Prior to completing the survey, participants were introduced to the study aims, study measures, and approximate time of survey completion. The survey form was restricted to accept only one response from the same IP address. After providing digital informed consent, participants were directed to the measures described in the Methods section. Participants who did not provide a consent to participate were acknowledged and the survey was terminated. Additionally, study participants could end the survey at any point by closing their browser, with their responses excluded.

The survey was implemented through the Google Forms web survey platform. All questions were set as mandatory, and all answers were selected from prepared lists of values. This avoided missing data, incorrectly completed surveys, and entry mistakes. The link to the anonymous survey was distributed in all main country municipalities using social networks of public health bureaus. In addition, information about the study was shown as a Facebook sponsored advertisement in the main cities of the country. The study was approved by the Social Research Ethics Board of Lithuanian Sports University (protocol number SMTEK-60, 24 November 2020).

### 2.2. Study Participants

Using the continuously varying sample size approach to Monte Carlo power analysis, approximately 150 individuals were required to ensure statistical power was at least 80% for detecting the hypothesized indirect effect [61]. For the multiple serial model, power of 0.80 can be achieved by a sample size *n* = 750. The calculated power for the sample size *n* = 900 was 0.88 (95% CI 0.84–0.91).

Nine persons did not consent to participate in the study. The final sample contained no missing data and consisted of 924 adults aged 18–79 years with a mean age of 40.0 ± 12.4 years. Of the total sample, 26.4% were men (*n* = 244) and the remaining 73.6% were women (*n* = 680). A more detailed sample description is provided in the Results section and Table 1.

### 2.3. Measures

Sociodemographic data were collected (gender identity, age, highest educational qualification, place of residence, marital status, financial security, ethnicity, height, and weight). All sociodemographic characteristics with their original response options are presented in Table 1. For this study, place of residence was classified into two groups: urban (capital, cities, towns) and rural (rural arears and suburbs). Height and weight data were used to compute self-reported BMI as kg/m^2^. Next, study variables are presented in a predetermined order.

The Nature Exposure Scale (NES) [62] contains four items asking participants to indicate how much of their routine environment is surrounded by nature (sample item: “Please rate the frequency (how often) of exposure to nature-rich environments outside of your everyday environment”) and the extent to which this environment is noticed (“How much notice would you take of the nature in these environments?”). Response options from 1 up to 5 were averaged and higher scores indicate greater nature exposure. The Lithuanian translation of the NES demonstrated adequate psychometric properties and unidimensional factor structure [63]. For this study, McDonald’s ω was 0.69 (95% CI = 0.64, 0.73).

The Body Appreciation Scale 2 (BAS-2) [22] was used to measure acceptance, respect, and care for one’s body and protection of one’s body from the internalization of sociocultural beauty standards. This unidimensional instrument contains 10 statements (sample item: “I feel good about my body”) with five response options from never up to always. An overall score was calculated by averaging response options. The Lithuanian translation of the BAS-2 demonstrated adequate psychometric properties and a unidimensional factor structure [64]. For this study, McDonald’s ω for scores on this scale was 0.96 (95% CI = 0.95, 0.96).

Frequency of physical activity in nature (PAN) was assessed using a single item: “How often do you exercise, go for a walk, cycle or do work demanding physical efforts in a natural environment (e.g., forest, park)?” Response options were provided in a frequency style as 1 = never or very rarely, 2 = 2–3 times a month, 3 = once a week, 4 = 2–4 times a week, 5 = 5–6 times a week, 6 = every day. This question was taken from a Lithuanian national survey and modified by asking respondents to indicate only PA in natural surroundings [65]. To avoid potential bias caused by different seasons, participants were asked to rate their frequency of PAN during the spring, summer, autumn, and winter. For analyses, mean PAN score across seasons was calculated. Correlations between seasonal frequencies of PAN varied from 0.63 to 0.86, and McDonald’s ω for overall scores was 0.95 (95% CI = 0.94, 0.95).

The Behavioral Regulation in Exercise Questionnaire 2 (BREQ-2) [63] is a 19-item, self-report instrument based on Self-Determination Theory (SDT) that assesses exercise motivations. The scale comprises five subscales assessing five types of exercise regulation: amotivation, external regulation, introjected regulation, identified regulation, and intrinsic motivation. Participants were asked to respond to items assessing why they engaged in physical activity and exercise and their responses were measured on a 5-point scale ranging from 1 (not true for me) to 5 (very true for me). The Lithuanian translation of the BREQ-2 has been shown to have the same 5-factor structure as the parent version, with adequate psychometric properties [47]. For this study, McDonald’s ω for the BREQ-2 subscales were as follows: for amotivation—0.82 (95% CI = 0.77, 0.85); external regulation—0.81 (95% CI = 0.78, 0.84); introjected regulation—0.68 (95% CI = 0.65, 0.73); identified regulation—0.76 (95% CI = 0.74, 0.79); and intrinsic motivation—0.91 (95% CI = 0.90, 0.93). For statistical analysis, we used the Relative Autonomy Index (RAI) calculated by the equation: (−3 × amotivation) + (−2 × external regulation) + (−1 × introjected regulation) + (2 × identified regulation) + (3 × intrinsic regulation), where higher scores indicate more autonomy in exercise regulation, whereas lower scores indicate more controlled regulation and/or amotivation [66].

The Restoration Outcome Scale (ROS) [67] measures positive restorative emotional and cognitive outcomes after contact with a natural environment (sample item: “I felt restored and relaxed”). The scale contains nine statements with a 7-point response scale ranging from 1 (not at all) to 7 (completely). An overall score was calculated by averaging response options. Previous work has shown that, in Lithuanian adults, ROS scores have a unidimensional factor structure with adequate internal consistency [68]. In the present study, McDonald’s ω for ROS scores was 0.97 (95% CI = 0.97, 0.98).

The Connectedness to Nature Scale (CNS) [49] is a widely used instrument that measures an individual’s affective and experiential connection to nature (sample item: “I think of the natural world as a community to which I belong.”). Fourteen items were rated on a 5-point scale from 1 (strongly disagree) up to 5 (strongly agree). An overall score is calculated as the mean of the response options. Scores on the CNS have been shown to have a unidimensional factor structure, with estimates supporting internal consistency and construct validity in Lithuanian adults [68]. In the present study, McDonald’s ω for the CNS was 0.90 (95% CI = 0.89, 0.91).

### 2.4. Statistical Analyses

First, normality testing of study variables was conducted. Study variables across urban and rural residence groups were compared by the independent samples *t*-test and effect sizes with Hedges’ *g* correction for different samples sizes were calculated. Correlations between study variables were tested by way of Pearson correlations. Internal consistency of the scales was tested by McDonald’s omega (ω) and presented with 95% CIs [69]. Preliminary statistical analysis was carried out using IBM SPSS Statistics v.28 (IBM Corp., Armonk, NY, USA).

Next, the study variables were included in mediational structural equation models. The bootstrap approach was used to conduct mediation analyses with 5000 bootstrap samples drawn from the dataset to calculate indirect and direct effects and bias corrected 95% CIs [70]. The 95% CIs for the coefficients calculated by bootstrapping methods were considered statistically significant if the confidence intervals did not include zero. Model fit was assessed using indices recommended by Hu and Bentler [71]: the normed model chi-square (χ^2^/df; values < 3.0 considered indicative of a good fit), the standardized root mean square residual (SRMR; values < 0.09 indicate a reasonable fit), the comparative fit index (CFI; values close to or > 0.95 indicate an adequate fit), and the root mean square error of approximation (RMSEA) and its 90% CI (values close to 0.06 indicative of good fit and values up to 0.08 indicative of adequate fit). Mediation analysis was conducted using AMOS v.26 (IBM Corp., Armonk, NY, USA).

## 3. Results

### 3.1. Characteristics of the Study Sample

In terms of educational qualifications, 11.3% of participants had completed secondary education or less, 7.7% were in full-time education, 41.7% had completed an undergraduate degree, and had a 34.8% postgraduate degree (4.4% of study respondents did not specify their educational attainment). Of the total sample, 15.5% resided in the country capital, 3.0% in capital suburbs, 34.4% in other cities, 20.0% in towns, and 27.1% in rural areas. Mean sample body mass index (BMI) was 24.8 ± 4.6 kg/m^2^ and ranged from 16.4 to 44.9 kg/m^2^. According to WHO criteria, all respondents were classified into groups of underweight (3.8%), normal weight (55.7%), overweight (28.3%), and obesity (12.2%). Sample characteristics are presented in Table 1.

Study variables were compared across urban and rural place of residence groups (Table 2). It was found that residents of rural areas reported a higher frequency of PAN with an effect size (ES) of 0.22, in addition to greater nature exposure (ES 0.37) (Table 1). Higher autonomous exercise motivation and body appreciation were observed in urban residents as compared to rural residents (ES 0.16 and 0.17 accordingly). There were no significant differences in perceived nature restoration and nature connectedness scores across urban and rural residents.

Comparing study variables across gender groups, only one significant difference was observed: mean PAN score was higher in men (3.69 ± 1.33) compared to women (3.44 ± 1.33, *p* = 0.012, ES = 0.19, not presented in a table).

### 3.2. Correlations between Study Variables

Inter-scale correlations are reported in Table 3. Scores on all variables were significantly and positively correlated with small-to-medium effect sizes.

### 3.3. Path Analysis

The hypothesized model did not demonstrate adequate model fit indices, χ^2^ = 28.416, *p* < 0.001; *df* = 2; CFI = 0.977; SRMR = 0.037; RMSEA = 0.112 (90% CI = 0.083, 0.160). Accordingly, non-significant paths were removed; specifically, we removed the pathways from nature exposure to body appreciation (estimate = −0.018, SE = 0.039, *p* = 0.651) and from physical activity in nature to connectedness to nature (estimate = 0.020, SE = 0.017, *p* = 0.245). The final model, presented in Figure 2, provided an adequate fit to the data, χ^2^ = 29.973, *p* < 0.001; *df* = 4; CFI = 0.977; SRMR = 0.039; RMSEA = 0.084 (90% CI = 0.057, 0.113). All retained path coefficients were significant with a positive valence.

Nature exposure was associated with more frequent PAN (estimate = 0.564, SE = 0.057, *p* < 0.001), whereas autonomy in exercise motivation was related to more frequent PAN (estimate = 0.039, SE = 0.006; *p* < 0.001). In addition, direct associations were found between nature exposure and connectedness to nature (estimate = 0.425, SE = 0.029, *p* < 0.001), nature exposure and nature restorativeness (estimate = 0.205, SE = 0.063, *p* = 0.001), and between nature connectedness and nature restorativeness (estimate = 1.056, SE = 0.062, *p* < 0.001). There were also significant direct effects from exercise autonomy to nature restorativeness (estimate = 0.017, SE = 0.006; *p* = 0.004) and body appreciation (estimate = 0.041, SE = 0.004; *p* < 0.001). Finally, direct associations were found between connectedness to nature and body appreciation (estimate = 0.166, SE = 0.040; *p* < 0.001), between nature restorativeness and body appreciation (estimate = 0.075, SE = 0.019; *p* < 0.001), and between frequency of PAN and body appreciation (estimate = 0.064, SE = 0.019, *p* < 0.001).

Table 4 describes indirect effects. There were significant serial mediations from nature exposure via PAN frequency, nature restorativeness, and connectedness to nature, to body appreciation. In addition, there was a significant serial mediation from nature exposure via nature connectedness to nature restoration effect. Finally, nature restoration mediated the association between exercise autonomy and positive body image.

Next, we assessed configural invariance of the final model across place of residence (urban and rural) and gender (men and women). Results showed that the model fitted the data across both women and men, χ^2^ = 33.010, *p* < 0.001; *df* = 8; CFI = 0.978; SRMR = 0.058; RMSEA = 0058 (90% CI = 0.038, 0.080), as well as urban and rural residents, χ^2^ = 36.855, *p* < 0.001; *df* = 8; CFI = 0.975; SRMR = 0.030; RMSEA = 0.063 (90% CI = 0.043, 0.084).

## 4. Discussion

In this study, we tested the mediating roles of physical activity in nature, connectedness to nature, perceived restorativeness, and autonomous motivation in explaining the associations between nature exposure and positive body image (operationalized as trait body appreciation). Based on previous findings, we predicted that physical activity in nature and autonomous motivation would mediate the association between nature exposure and positive body image. We also predicted that nature restoration would mediate relationships between nature exposure and positive body image on the one hand and connectedness to nature on the other hand.

### 4.1. Physical Activity in Nature and Physical Activity Motivation as the Mediators between Nature Exposure and Positive Body Image

First, our results showed that nature exposure was indirectly associated with positive body image via physical activity in nature and autonomous motivation. The mediating role of physical activity in the associations between nature exposure and well-being has been observed in previous studies [5,32]. Previous work has also suggested that visiting natural environments promotes physical activity especially in leisure-time and mitigates feelings of loneliness [32]. One possible explanation of these results is based on the theoretical framework of ecological dynamics, which states that physical activity in nature (especially active transport, such as walking and cycling) and leisure recreation (taking part in various sports and recreational activities) are likely associated with a greater sense of variability of nature and demands the holistic (cognitive and emotional) involvement of an individual [34]. Exercising in constantly changing natural environments captures the attention of the exerciser and requires interaction with natural challenges, while possibly decreasing the exercisers’ stress and negative body image-related emotions and body surveillance. Another possible explanation is based on the theory of positive embodiment [29]. It may be assumed that physical activity in nature, especially exercising, provides an additional indirect positive body image-enhancing effect through increased feelings of body functionality [30], because the motivation of physical activity in nature is of a more intrinsic nature, i.e., more body functionality, but is not body image oriented [46,58]. Further, outdoors exercising is associated with lower somatic anxiety [59].

The results of the present study also suggested that intrinsic motivation is a mediator of the relationship between nature exposure and frequency of physical activity on the one hand and nature exposure and positive body image on the other hand. These findings are in line with the main tenets of SDT theory, which suggests that intrinsic motivation is associated with higher physical activity and more positive body image, because it helps to fulfil three basic human needs, namely ARC [38,72].

Finally, the associations between nature exposure and greater body appreciation can be explained by elevated mindfulness when exercising or being physically active in nature. Recent research exploring mindfulness in physical activity has suggested that state mindfulness in physical activity is associated with lower body surveillance, greater mood enjoyment, more autonomous exercise motivation, and more internal exercise goals and body appreciation [73,74]. It may be that physical activity in nature is more mindful compared to exercising in indoors and mindfulness in physical activity may mediate associations between nature exposure, physical activity in nature, and body appreciation. Therefore, future studies should test this assumption.

### 4.2. Restoration in Nature and Connectedness to Nature as Mediators between Nature Exposure and Positive Body Image

In the present study, we found that perceived restoration and connectedness to nature were mediators between nature exposure and positive body image. Thus, our findings replicate previous results in Western European samples suggesting that nature connectedness is directly associated with positive body image [16] and mediates the associations between nature exposure and positive body image [19]. However, the novel contribution of the present research is that perceived restoration mediated the association between nature exposure and positive body image. To our knowledge, this is one of the first studies testing the nature restoration effect in this association. These results may be explained by SRT [13] and ART [14,15]. Urbanized environments are highly appearance-oriented, whereas being in nature may reduce stress related to body image concerns and enhance positive emotions, helping people to rest from cognitive fatigue related to sociocultural pressures towards stereotyped body appearance [26]. However, the present study is cross-sectional and future studies with experimental designs should confirm our findings.

### 4.3. Differences in Nature Exposure, Body Appreciation, Physical Activity in Nature, Physical Activity Motivation, Nature Connectedness, and Restorativeness in Urban and Rural Residents

We assessed differences in study variables across urban and rural inhabitants. Respondents living in rural areas reported greater nature exposure. Previous studies demonstrated that increasing urbanization is associated with a decrease in the frequency, duration, and intensity of nature exposure [75]. Further, respondents living in natural surroundings reported greater physical activity in nature. This finding is in line with other findings demonstrating higher nature physical activity in people having more green spaces in their place of residence [76,77,78]. Next, we observed no differences in nature connectedness and perceived restorativeness between urban and rural residents. The present study showed that urban residents demonstrated greater body appreciation. This finding contradicts findings of previous studies demonstrating greater positive body image in rural Malaysian women [79]. However, the studies exploring positive body image in terms of place of residence are scarce; therefore, these findings should be interpreted with caution and future studies are recommended.

### 4.4. Practical Implications

The findings of the present study have important implications for practice. Including self-determined physical activity in nature may increase the effectiveness of universal intervention programs aiming to strengthen positive body image. Our findings suggest that exercising in natural environments may be more beneficial for the development of positive body image, because it provides more restoration effects through higher environment variability compared to traditional exercising indoor which is usually highly body image oriented [33,80]. Moreover, physical activity in nature may more effectively strengthen body functionality compared to indoor exercising because exercising in nature requires more attention to continuously changing environmental surroundings. Strengthening connectedness to nature may be also a useful way for intervention programs to promote healthier body image. However, this study remains preliminary and future studies should provide more empirical data on this issue. Future studies of experimental design should test our findings including the changes in body functionality and mindfulness in the associations between nature exposure and positive body image.

### 4.5. Strengths and Limitations of the Present Study

The strength of the present study is that associations between nature exposure and positive body image were tested in a large sample of adults across gender and various ages. The present study is also one of the first attempts to replicate previous findings of Western European countries in Eastern Europe. Nevertheless, the present study has important limitations that should be discussed. First, the study is cross-sectional and the associations between the variables cannot be considered as causal. However, previous experimental and pseudo-experimental studies on the effect of nature exposure on positive body image lead us to believe that regular exposure to natural environments increases positive body image, but not vice versa [10,18,25,26]. Another limitation of the present study is that we assessed positive body image as body appreciation. However, positive body image is a multifaceted construct [21] and it would be useful to re-examine our findings with additional facets of positive body image. Further, only a quarter of the sample were men; thus, generalization of the findings should be limited and tested by future studies. Finally, we did not screen for mental health conditions in this sample, which may be a limitation as some of the constructs measured here may differ in epistemological or phenomenological meaning between those with and without mental health conditions. This is an issue that may be addressed more fully in future studies by examining the factorial invariance of our modelling across such groups.

## 5. Conclusions

The findings of the present study suggest that physical activity in nature mediates the relationship between nature exposure and body appreciation. Autonomy in exercise, connectedness to nature, and perceived restoration directly and indirectly affected associations between nature exposure and body appreciation. Enhancing nature exposure, increasing physical activity in nature, and strengthening autonomous physical activity motivation may be an effective strategy in interventions promoting positive body image.

## Figures and Tables

**Figure 1 ijerph-18-12246-f001:**
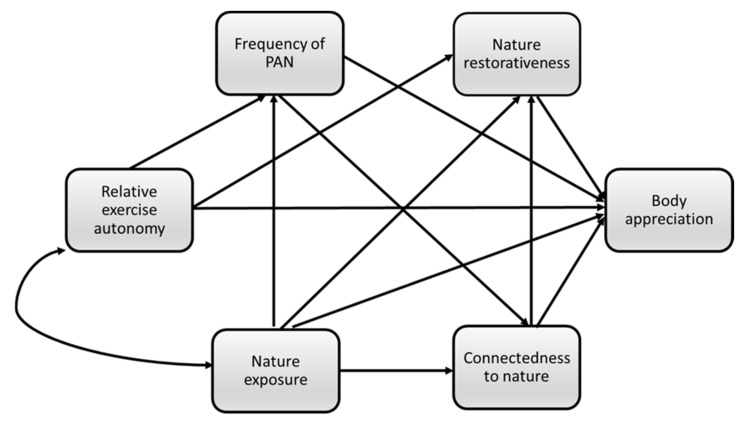
Hypothesized model of the associations between study variables. PAN = physical activity in nature.

**Figure 2 ijerph-18-12246-f002:**
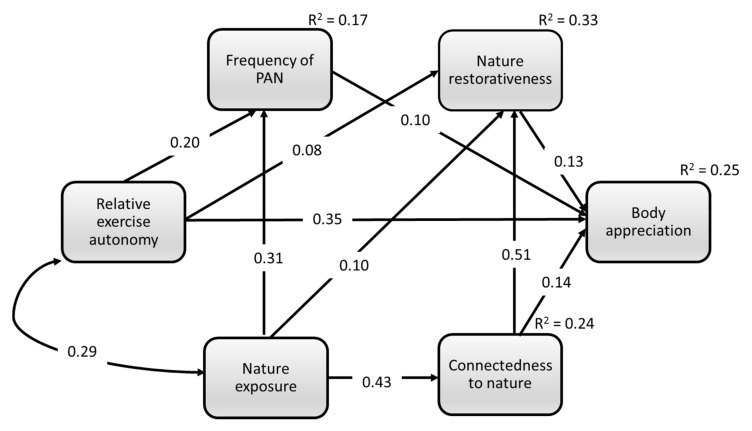
The final path model of the association between nature exposure and positive body image mediated by physical activity in nature (PAN), nature restorativeness and connectedness to nature with standardized estimates (*n* = 924). Note. All regression coefficients are significant (*p* < 0.01).

**Table 1 ijerph-18-12246-t001:** Sample characteristics (*n* = 924).

Characteristics		*n*	%
Gender	men	244	26.4
	women	680	73.6
Age, years (m ± SD) 40.0 ± 12.4			
Education	secondary	105	11.3
	in full time studies	71	7.7
	undergraduate degree	385	41.7
	postgraduate degree	322	34.8
	other	41	4.5
Marital status	single	170	18.4
	single but in a committed relationship	159	17.2
	married	534	57.8
	other	61	6.6
Place of residence	capital city	143	15.5
	capital suburb	28	3.0
	provincial city with more than 100,000 inhabitants	318	34.4
	provincial town with more than 10,000 inhabitants	185	20.0
	rural area	250	27.1
Ethnicity	ethnic majority	842	91.1
	ethnic minority	27	2.9
	not sure	55	6.0
Financial security	less secure compared to others	158	17.1
	same	599	64.8
	more secure compared to others	167	18.1
Body mass index, kg/m^2^	underweight (<18.5 kg/m^2^)	35	3.8
	normal weight (18.5–24.9 kg/m^2^)	514	55.7
	overweight (25.0–29.9 kg/m^2^)	261	28.3
	obesity (≥30.0 kg/m^2^)	112	12.2

m = mean, SD = standard deviation.

**Table 2 ijerph-18-12246-t002:** Comparison of the study variables (m ± SD) across urban vs. rural place of residence groups (*n* = 924).

Variables	Urban *n* = 646	Rural *n* = 278	*t*	Cohen’s *d*	*p*
Nature exposure	4.02 ± 0.72	4.28 ± 0.72	−5.12	−0.37	<0.001
Relative exercise autonomy	9.25 ± 6.74	8.15 ± 7.30	2.16	0.16	0.031
Frequency of PAN	3.42 ± 1.30	3.72 ± 1.39	−3.11	−0.22	0.002
Nature restorativeness	5.40 ± 1.47	5.40 ± 1.57	0.01	0.001	0.99
Connectedness to nature	3.80 ± 0.71	3.84 ± 0.73	−0.086	−0.06	0.39
Body appreciation	3.75 ± 0.80	3.61 ± 0.92	2.18	0.17	0.03

PAN = physical activity in nature, m = mean, SD = standard deviation, CI = confidence interval, *p* = significance level.

**Table 3 ijerph-18-12246-t003:** Correlations between study variables (*n* = 924).

Variables	NES	RAI	PAN	ROS	CNS	BAS-2
Nature Exposure Scale (NES)	1					
Exercise Autonomy Index (RAI)	0.29 **	1				
Frequency of Physical Activity in Nature (PAN)	0.37 **	0.29 **	1			
Restoration Outcomes Scale (ROS)	0.34 **	0.25 **	0.11 **	1		
Connectedness to Nature Scale (CNS)	0.43 **	0.27 **	0.19 **	0.57 **	1	
Body Appreciation Scale 2 (BAS-2)	0.23 **	0.44 **	0.24 **	0.31 **	0.33 **	1

** *p* < 0.01.

**Table 4 ijerph-18-12246-t004:** Summary of mediation analyses testing the indirect effect between study variables (*n* = 924).

Paths	β (95% CI)	*p*
Nature exposure→connectedness to nature→nature restorativeness	0.219 (0.181, 0.258)	0.001
RAI→nature restorativeness→body appreciation	0.032 (0.015, 0.053)	0.001
Nature exposure→body appreciation (via PAN frequency, nature connectedness and nature restorativeness)	0.137 (0.102, 0.171)	0.001
Connestedness to nature→nature restorativeness→body appreciation	0.068 (0.033, 0.109)	0.001

RAI = relative autonomy index, PAN = physical activity in nature, β = standardized effect coefficient, 95% CI = 95% confidence intervals for standardized effect, *p* = two-tailed significance.

## Data Availability

The dataset generated and analyzed during the current study is available from the corresponding author upon reasonable request.

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
