# Peer review of "Nature Exposure and Positive Body Image: A Cross–Sectional Study Examining the Mediating Roles of Physical Activity, Autonomous Motivation, Connectedness to Nature, and Perceived Restorativeness"

_ijerph, 2021, doi:10.3390/ijerph182212246_

Round 1

Reviewer 1 Report

Reviewer's comments on the manuscript entitled Nature exposure and positive body image: Examining the mediating roles of physical activity, autonomous motivation, connectedness to nature, and perceived restorativeness” (ijerph-1412338). The aim of the study was to test the mediating roles of physical activity in nature, perceived restoration in nature, autonomous motivation, and connectedness to nature in explaining the association between nature exposure and positive body image.

I. General comments on the manuscript:

1. The manuscript should be prepared by its authors in accordance with the editorial standards and requirements.

2. There is no need to separate parts in the summary.

3. It is important to prepare the text in a way that there are separate paragraphs and no spaces between individual parts in the chapter.

4. Tables should be prepared in accordance with the instructions and guidelines for authors.

5. The literature on the subject needs to be prepared in a way that it meets editorial requirements.

II. Introduction

6. The authors of the paper should take shortening the introduction by half into consideration.

III. Materials and Methods

7. It is important to provide a detailed description of the research design in subsection 2.1. Please answer the following questions: What was the name of a web portal with the survey questionnaire available? How did the respondents give their informed consent to take part in the survey? What happened, if someone did not give his or her informed consent? Could the respondent stop the survey? In what way? Where did the respondents come from? Did potential research participants have the same access to the survey questionnaire? etc. There are many issues that need to be clarified and therefore, I would like to ask the authors to do so.

8. In subsection 2.2, please answer the following questions in detail: What were the inclusion and exclusion criteria for the study group?How many survey questionnaires were excluded from the study and what was the reason? Did the authors take survey questionnaires completed, for instance too soon, into consideration? I suggest presenting a figure indicating how the study sample was selected. In addition, I recommend removing the description of the characteristics of the study group from this subsection and transfer it to the “Results” part and start presenting the findings from them. A table with the characteristics of the study group may also be created. This will certainly make the paper clear to the reader.

9. There is chaos in subsection 2.3. The authors of the manuscript need to put this important part in order. I suggest starting this subsection by providing information that five standardised tools and two author's own tools were used in this study. It is important to describe the standardised tools at the very beginning. Then, the authors should describe the scope of the survey questionnaire assessing physical activity in nature (PAN) and demographics. I recommend describing the tools by using references in order to put this part of work in order. While describing the standardised tool, there is no need to cite sample questions related to this tool.

IV. Results

10. In this part of the study, I suggest introducing subchapters to make the work more reader-friendly.

11. Please, start this part of the paper by describing the characteristics of the study group. Please, take making a table into consideration.

V. Discussion

12. Like the introduction, the discussion part is extensive. The authors of the paper should take shortening the introduction in favour of the discussion into consideration. Certain information included in the introduction is repeated in the discussion. What is the purpose of reading the same thing twice?

VI. Conclusions

13. Using full names instead of abbreviations is highly recommended.

Author Response

Dear Reviewer,

Thank you for your time reviewing our paper and for your valuable comments. All changes made in the text are highlighted in a blue font.

Reviewer's comments on the manuscript entitled “Nature exposure and positive body image: Examining the mediating roles of physical activity, autonomous motivation, connectedness to nature, and perceived restorativeness” (ijerph-1412338). The aim of the study was to test the mediating roles of physical activity in nature, perceived restoration in nature, autonomous motivation, and connectedness to nature in explaining the association between nature exposure and positive body image.

  1. General comments on the manuscript:
  2. The manuscript should be prepared by its authors in accordance with the editorial standards and requirements.

The manuscript was double-checked and revised according the requirements.

  1. There is no need to separate parts in the summary.

Accepted and revised.

  1. It is important to prepare the text in a way that there are separate paragraphs and no spaces between individual parts in the chapter.

Accepted and revised. Formating will be double-checked by a layout designer at the final stage.

  1. Tables should be prepared in accordance with the instructions and guidelines for authors.

Accepted and revised.

  1. The literature on the subject needs to be prepared in a way that it meets editorial requirements.

Accepted and revised.

  1. Introduction
  2. The authors of the paper should take shortening the introduction by half into consideration.

Reviewer No.3 also mentioned that introduction is quite long for the original paper however, however, reviewer recommended maintaining the introduction since it is „interesting and well written“. We double-checked the introduction and revised some parts, however, the major body of the text was left as it is. Importantly, there was no recommendation to shorten introduction from the reviewer No. 2.

III. Materials and Methods

  1. It is important to provide a detailed description of the research design in subsection 2.1. Please answer the following questions: What was the name of a web portal with the survey questionnaire available? How did the respondents give their informed consent to take part in the survey? What happened, if someone did not give his or her informed consent? Could the respondent stop the survey? In what way? Where did the respondents come from? Did potential research participants have the same access to the survey questionnaire? etc. There are many issues that need to be clarified and therefore, I would like to ask the authors to do so.

All these questions were answered and the information was inserted in subsection 2.1 (Study procedure).

  1. In subsection 2.2, please answer the following questions in detail: What were the inclusion and exclusion criteria for the study group? How many survey questionnaires were excluded from the study and what was the reason? Did the authors take survey questionnaires completed, for instance too soon, into consideration? I suggest presenting a figure indicating how the study sample was selected. In addition, I recommend removing the description of the characteristics of the study group from this subsection and transfer it to the “Results” part and start presenting the findings from them. A table with the characteristics of the study group may also be created. This will certainly make the paper clear to the reader.

All these questions were answered and information was inserted in subsection 2.1. A detailed reqruitment of the study participants and inclusion criteria are described in the section 2.1. We moved sample description to the results section and added Table 1 with the sample characteristics as recommended.

  1. There is chaos in subsection 2.3. The authors of the manuscript need to put this important part in order. I suggest starting this subsection by providing information that five standardised tools and two author's own tools were used in this study. It is important to describe the standardised tools at the very beginning. Then, the authors should describe the scope of the survey questionnaire assessing physical activity in nature (PAN) and demographics. I recommend describing the tools by using references in order to put this part of work in order. While describing the standardised tool, there is no need to cite sample questions related to this tool.

According to the suggestion, we revised subsection 2.3. and changed the order of instruments starting from sociodemographics and following the order of instruments‘ by their importance in the study.  All instruments are validated official scales or measures and non of them are self-developed.

First, we presented the main instruments (nature exposure and positive body image (body appreciation-2) scales. Next, we presented mediators – 1) physical activity in nature questionnaire and behavioral regulation in exercise questionnaire – 2, and 3) connectedness to nature scale and restorative outcome scale.

  1. Results
  2. In this part of the study, I suggest introducing subchapters to make the work more reader-friendly.

Subchapters were  included in the results section.

  1. Please, start this part of the paper by describing the characteristics of the study group. Please, take making a table into consideration.

A table with the sample characteristics is included as recommended.

  1. Discussion
  2. Like the introduction, the discussion part is extensive. The authors of the paper should take shortening the introduction in favour of the discussion into consideration. Certain information included in the introduction is repeated in the discussion. What is the purpose of reading the same thing twice?

We revised discussion section and excluded information that was repeated.

  1. Conclusions
  2. Using full names instead of abbreviations is highly recommended.

Conclusions were double-checked and full names insted of abbreviations were included.

Reviewer 2 Report

Overall, I think this is an interesting study, and well presented, and of interest to readers. I have a few comments:

  1. Title: I think this needs to represent that this is a survey-based cross-sectional study so that the reader understands what kind of study it is. I thought this was a qualitative study on reading the title.
  2. Abstract: Methods: I think the measures should be stated. The results are not displayed: I think that coefficient and CI for the key associations should be shown, and the role of PAN in mediating the relationship between NE and body appreciation put into the conclusion/discussion.
  3. Results: I would like to see a table showing the breakdown in study participants eg. male/female/BMI/ ages/ and average scores over the variables measured. I would want to know how representative the sample is.
  4. Discussion: I think the limitations are well presented, but I also think that there needs to be some emphasis on the implications of these findings in practice. Might PAN be a treatment option for those with low body appreciation for example?

Author Response

Review 2

Overall, I think this is an interesting study, and well presented, and of interest to readers. I have a few comments:

Title: I think this needs to represent that this is a survey-based cross-sectional study so that the reader understands what kind of study it is. I thought this was a qualitative study on reading the title.

Thank you for the postive comment and for your time reviewing our paper. The name of the study was revised according to the remark. All changes made in the text are highlighted in a blue font. Thank you once again.

Abstract: Methods: I think the measures should be stated. The results are not displayed: I think that coefficient and CI for the key associations should be shown, and the role of PAN in mediating the relationship between NE and body appreciation put into the conclusion/discussion.

Abstract was revised according to the recommendations.

Results: I would like to see a table showing the breakdown in study participants eg. male/female/BMI/ ages/ and average scores over the variables measured. I would want to know how representative the sample is.

Informations was included as Table 1.

Discussion: I think the limitations are well presented, but I also think that there needs to be some emphasis on the implications of these findings in practice. Might PAN be a treatment option for those with low body appreciation for example?

Practical implications were included  at the end of the discussion section as recommended.

Reviewer 3 Report

This is a very nice study addressing an interesting topic.

There is a very large literature review section in the beginning, which is very informative, but actually beyond what would be expected for an original article. I wonder if the article should be called something like “review of current evidence and….”. I would not remove the long intro since it is interesting and well written.

I have two main concerns:

  • To me it remains unclear how exactly the sample was drawn (….public health bureaus) but his is very vague. I am surprised that I would no information on mental and physical disorders present in these individuals since this could have influenced the results. Please address this issue. Were these all healthy individuals? Mental disorders affect almost 30% of the population, how did you screen for this?
  • I get the impression that the term “nature” is used interchangeably with “blue and green spaces” in this study which poses some problems. “nature” in a more narrow sense is a not a park or a lawn but natural surroundings not heavily influenced by human influence. So I think I would make sense to distinguish this.

Author Response

Review 3

This is a very nice study addressing an interesting topic.

There is a very large literature review section in the beginning, which is very informative, but actually beyond what would be expected for an original article. I wonder if the article should be called something like “review of current evidence and….”. I would not remove the long intro since it is interesting and well written.

Thank you for the positive comment and for your time reviewing our paper. All changes made are highlighted in the text in a blue font.

I have two main concerns:

To me it remains unclear how exactly the sample was drawn (….public health bureaus) but his is very vague.

The information about development of the sample was expanded.

I am surprised that I would no information on mental and physical disorders present in these individuals since this could have influenced the results. Please address this issue. Were these all healthy individuals? Mental disorders affect almost 30% of the population, how did you screen for this?

We asume that the sample of the present study were healthy people of various ages. There were no questions about previous mental ilnesses included. We will take this remark into consideration in future studies.

I get the impression that the term “nature” is used interchangeably with “blue and green spaces” in this study which poses some problems. “nature” in a more narrow sense is a not a park or a lawn but natural surroundings not heavily influenced by human influence. So I think I would make sense to distinguish this.

We use term „nature exposure“ in all text and excluded „exposure to blue and green spaces“ from two places in text by changing them to „nature exposure“.

Round 2

Reviewer 3 Report

I find that two of my comments are not sufficiently adressed:

1) I could still not detect a definition of "nature" in the text. I think this is essential since "nature" can be either an urban balcony with two flower pots or a national park......quite a different experience. I think this should be defined clearer throughout the text.

2) The authors state that they did not screen for any disorders (neither mental nor physical). Especially the lack of screening for mental disorders is a strong limitation in my opinion and should be adequately adressed in the manuscript. Many of the constructs the authors measure probably differ between mentally healthy individuals and those with a mental disorder and it is unclear wether the described associations are to be found in both populations.

Author Response

Dear Reviewer,

On behalf of all authors, I wish to thank you for your time and input during the peer review process.

We revised the paper according to reviewer's comments. Changes made are highlighted in yellow.